# Immunotherapy-Related Hypophysitis: A Narrative Review

**DOI:** 10.3390/cancers17030436

**Published:** 2025-01-27

**Authors:** Vincenza Di Stasi, Domenico La Sala, Renato Cozzi, Francesco Scavuzzo, Vincenzo De Geronimo, Maurizio Poggi, Mario Vitale, Anna Tortora

**Affiliations:** 1Center of Nutrition for the Research and the Care of Obesity and Metabolic Diseases, National Institute of Gastroentherology IRCCS Saverio De Bellis, 70013 Castellana Grotte, Italy; vincenza.distasi@irccsdebellis.it; 2UOSD Malattie Endocrine Nutrizione e Ricambio, AORN, San Giuseppe Moscati, 83100 Avellino, Italy; 3Endocrine Unit Grande Ospedale Metropolitano, Niguarda, 20162 Milano, Italy; dr.renatocozzi@gmail.com; 4UOC Endocrinologia AORN A. Cardarelli, 80131 Napoli, Italy; francesco.scavuzzo@aocardarelli.it; 5Centro Clinico Diagnostico GB Morgagni srl, 95125 Catania, Italy; vdg@iol.it; 6UOC Medicina Specialistica Endocrino-Metabolica, AOU Sant’Andrea, 00189 Roma, Italy; mpoggi@ospedalesantandrea.it; 7Dipartimento di Medicina, Chirurgia e Odontoiatria, Università di Salerno, 84081 Baronissi, Italy; mavitale@unisa.it; 8UOC Clinica Endocrinologica e Diabetologica, AOU San Giovanni di Dio e Ruggi d’Aragona, 84131 Salerno, Italy

**Keywords:** immune checkpoint inhibitors, hypophysitis, immunotherapy

## Abstract

Immune checkpoint inhibitors (ICIs) have transformed cancer therapy but are associated with immune-related side effects, including hypophysitis, which can significantly impair endocrine function. This review delves into the pathogenesis, clinical manifestations, diagnosis, and management of ICI-induced hypophysitis, emphasizing its diagnostic challenges due to symptom overlap with the underlying cancer. Accurate diagnosis requires a combination of clinical assessment, hormonal evaluation, and imaging studies. Management strategies center on tailored hormone replacement therapy and close monitoring. Early detection and prompt intervention are essential to enhancing patient outcomes and quality of life. This review aims to improve healthcare providers’ understanding and awareness of this complex and demanding condition.

## 1. Introduction

The discovery of immune checkpoint mechanisms has been instrumental in the development of specific inhibitors, known as immune checkpoint inhibitors (ICIs) [1]. ICIs are monoclonal antibodies designed to block immune tolerance and enhance immune surveillance against tumors [2]. This novel class of drugs bolsters the patient’s immune system, enabling it to better recognize and combat a growing range of cancers [3], including melanoma, non-small-cell lung cancer, renal cell carcinoma, prostate cancer, Hodgkin lymphoma, and head and neck cancers.

## 2. Overview of Basic Principles of T-Cell Immune Response to Antigen

Hematopoietic stem cells in the bone marrow give rise to T-cells, which migrate to and mature in the thymus. Mature T-cells express the T-cell receptor (TCR) on their surface, a unique antigen-binding receptor. Each T-cell displays a single type of TCR and, upon receiving the appropriate signal, undergoes rapid proliferation and differentiation.

T-cell activation begins when the TCR binds to a specific antigen presented by the major histocompatibility complex (MHC) on antigen-presenting cells (APCs).

APCs include dendritic cells, macrophages, B-cells, fibroblasts, and epithelial cells. The MHC consists of membrane proteins expressed on nucleated cells (MHC class I, also known as HLA-A, HLA-B, and HLA-C) or a subset of cells such as macrophages, dendritic cells, and B-cells (MHC class II).

Class I MHC presents endogenous peptides (synthesized within the cell) to T-cells, whereas class II MHC presents exogenous peptides that have been ingested and processed proteolytically [4,5].

While peptide recognition by the TCR is the first essential signal for T-cell activation, it is not sufficient. A secondary, costimulatory signal is required, provided by the interaction between the T-cell-expressed CD28 receptor and the CD80 or CD86 ligands (members of the B7 family) on activated APCs. This interaction promotes T-cell survival and proliferation through enhanced glucose metabolism and the expression of anti-apoptotic proteins like Bcl-X.

T-cells also express two immunomodulatory membrane proteins: cytotoxic T-lymphocyte-associated antigen 4 (CTLA4) and programmed cell death protein 1 (PD1).

CTLA4, a member of the immunoglobulin superfamily, is constitutively expressed in regulatory T-cells. Its activation increases proliferation and the production of immunosuppressive cytokines, such as IL-10 and TGF-β, as well as immunosuppressive molecules like indoleamine 2,3-dioxygenase.

In effector T-cells, CTLA4 resides in cytosolic vesicles beneath the plasma membrane in resting conditions.

Following T-cell activation, CTLA4 is externalized, where it competes with CD28 for binding to CD80/CD86. This interaction disrupts the APC cytoskeleton, weakens the TCR-APC connection, promotes internalization of CD80/CD86, and prevents their interaction with CD28. These processes downregulate T-cell survival and proliferation pathways, ultimately dampening the immune response [6,7,8,9,10].

Beyond lymphocytes, CTLA4 expression has been detected in various tissues, including the lungs, gastric mucosa, enteroendocrine cells, gallbladder, urinary bladder, breast, placenta, skin, thyroid, testes, and pituitary gland (RNA-seq datasets at https://www.proteinatlas.org, (URL accessed on 28 November 2024) and Genotype-Tissue Expression RNA-seq dataset at https://www.gtexportal.org, URL accessed on 28 November 2024) [11,12].

Programmed cell death protein 1 (PD1), a transmembrane glycoprotein of the immunoglobulin superfamily, is expressed on activated T-cells, B-cells, monocytes, and dendritic cells. PD1 expression has also been observed in non-hematopoietic tissues, including the testes, cerebral cortex, and pituitary gland (RNA-seq dataset at https://www.proteinatlas.org/, URL accessed on 28 November 2024) [12].

The pituitary gland has been identified as the sixth most significant source of PD1 expression in humans, according to the GTEx database and human protein atlas (RNA-seq dataset at https://www.proteinatlas.org/, URL accessed on 28 November 2024 and Genotype-Tissue Expression RNA-seq dataset at https://www.gtexportal.org, URL accessed on 28 November 2024) [12]; however, there is no unanimous consensus in the literature on this finding [13,14,15].

PD1 binds to its ligands, PDL1 (also known as B7-H1) and PDL2 (B7-DC), which are constitutively expressed on the surface of APCs. These ligands can also be induced by pro-inflammatory stimuli in non-hematopoietic cells and cancer cells. Notably, PDL1 is expressed constitutively in the hypothalamus.

When PD1 engages its ligands, it induces apoptosis in T-cells or drives them into a dysfunctional state known as T-cell exhaustion. This process is mediated through the recruitment of the tyrosine phosphatase SHP2 (also called PTPN11), which inhibits T-cell signaling.

CTLA4 and PD1 both attenuate immune responses but through distinct spatial and temporal mechanisms. CTLA4 acts early, through intrinsic and extrinsic pathways, while PD1 functions later, primarily via intrinsic and peripheral mechanisms [12,16,17].

When CD28-CD80/86 signaling predominates, T-cells become activated and differentiate into two primary subtypes: CD8+ T-cells and CD4+ T-cells.

CD8+ T-cells are activated through interactions with MHC class I molecules on APCs. These cells play a cytotoxic role, targeting infected or tumor cells by releasing substances that induce apoptosis.

CD4+ T-cells lack cytotoxic and phagocytic properties but are crucial for recruiting and coordinating other immune components. CD4+ T-cells are further categorized into regulatory T-cells and effector T-cells.

Regulatory T-cells (Treg CD4+CD25+) constitutively express CTLA4 and have an immunosuppressive function, modulating the immune response to prevent excessive activity.

Effector T-cells are divided into three major subtypes, each defined by distinct cytokine profiles: Th1 cells produce IFN-γ, TNF-α, and IL-2. They support cell-mediated immunity by activating macrophages, stimulating immune responses against intracellular pathogens, and promoting B-cell differentiation; Th2 cells release IL-4, IL-5, and IL-13, driving humoral immunity. They support the development of IgE-producing B-cells and recruit mast cells and eosinophils; Th17 cells, recently characterized, produce cytokines from the IL-17 family. They are associated with chronic inflammation and persistent infections [4,5,7,10].

Circulating naive effector T-cells can follow three primary fates upon activation. First, as the immune response subsides, or in cases of overstimulation, they undergo apoptosis. Second, T-cells may adopt an exhausted phenotype, a condition triggered by repeated low-dose and low-affinity stimulation, as observed in chronic infections and neoplastic processes. Third, a subset of these cells contributes to long-term immunological memory and is referred to as memory T-cells [4,9].

Figure 1 and Figure 2 below illustrate the main concepts discussed above.

## 3. Immune Checkpoint Inhibitors: Mechanism of Action

Many cancers possess an inherent ability to evade immune surveillance by modulating immune checkpoints. Tumor cells expressing CD80/86 can induce the translocation of CTLA4 from the cytoplasm to the cell surface in tumor-infiltrating lymphocytes, resulting in the suppression of T lymphocyte activity. Similarly, when PD-L1 or PD-L2 expressed on tumor cells binds to the PD-1 receptor on activated T-cells, a comparable suppressive effect occurs.

The therapeutic principle of ICIs is to reverse cancer-induced inhibition of the T-cell response, as illustrated in Figure 3 [18,19,20]. The most effective ICIs include ipilimumab, which targets CTLA4, pembrolizumab and nivolumab, which target PD-1, and atezolizumab, avelumab, and durvalumab, which target the PD-1 ligand (PD-L1) [21].

Ipilimumab became the first FDA-approved ICI in 2011 for the treatment of advanced melanoma. Clinical trials demonstrated its effectiveness in reducing mortality among patients with advanced-stage melanoma, inspiring the development of additional drugs targeting the same pathway [22]. However, the efficacy of anti-CTLA4 inhibitors has shown significant limitations. For instance, tremelimumab, developed following the success of ipilimumab, was denied FDA approval as a monotherapy [4]. Additionally, ipilimumab exhibited limited clinical benefit for patients with prostate cancer or small-cell and non-small-cell lung cancers [23].

This led researchers to focus on other immune checkpoints. The second generation of ICIs targets the PD-1/PD-L1 pathway. PD-1 inhibitors such as pembrolizumab and nivolumab have significantly extended the survival of patients with various malignancies, including non-small-cell lung cancer and colorectal cancer, where anti-CTLA4 inhibitors were less effective [3].

In recent years, the FDA has approved additional ICIs, with several clinical trials demonstrating their efficacy even in patients previously considered untreatable [24,25]. Notably, on 18 March 2022, the FDA approved the combination of two immunotherapy drugs, nivolumab and relatlimab, for treating adults and children aged 12 and older with advanced melanoma. These drugs target PD-1 and Lymphocyte Activation Gene 3 (LAG-3) on T lymphocytes, respectively. The approval followed a phase II/III trial which showed a significant progression-free survival benefit with the combination therapy compared to nivolumab monotherapy [26].

Thus, ICIs can be administered as monotherapies or in combination with other agents, including dual immune checkpoint blockades.

Table 1 outlines the FDA-approved ICIs and their current clinical indications.

Another target of immunotherapy is LAG-3, expressed on activated T-cells, B-cells, NK cells, and plasmacytoid dendritic cells, which interacts with major histocompatibility complex II (MHCII) with higher affinity than CD4. It downregulates T-cell proliferation, activation, and homeostasis [42,43] and promotes Treg cells suppression function [44] using a similar mechanism to CTLA4 or PD1 receptors.

Monoclonal antibody relatlimab acts on the expression of the LAG3 gene; in combination with nivolumab, an antibody directed against PD-1, it has given satisfying results in terms of overall survival in patients with advanced stage melanoma.

## 4. Immunotherapy-Related Hypophysitis

Infiltration of CD4+ T-cells with a Th1/Th17 cytokine profile, along with B lymphocytes producing anti-pituitary antibodies, has been documented in a mouse model of autoimmune hypophysitis induced by subcutaneous injection of pituitary extract, either alone or emulsified with complete Freund’s adjuvant (CFA). This suggests the involvement of both cellular and humoral immunity [45].

Transcriptomic and proteomic analyses also detected the presence of INFɤ and IL-17A. However, the mechanisms underlying immunotherapy-induced hypophysitis (IH) remain unclear, with some studies producing conflicting findings. As seen in other autoimmune conditions, a subset of Th17 cells can transition to a Th1 phenotype while retaining some Th17 features, potentially resulting in more aggressive behavior. However, a study analyzing human pituitary tissue from patients with hypophysitis (including three with IH) did not confirm this result. In that study, only elevated transcript levels of IL-17A were detected, without corresponding increases in INFɤ [45].

It has been hypothesized that the tissue analyzed could represent an early stage of inflammation where Th17 cells are predominant, while the switch to other lymphocyte phenotypes, such as Th1, may occur in later stages, consistent with observations in other autoimmune diseases [45,46].

Recently, the guanine nucleotide-binding protein G(olf) subunit alpha (GNAL) and the integral membrane protein 2B (ITM2B) have been identified as targets of autoantibodies in hypophysitis [47,48].

GNAL, primarily expressed in the olfactory epithelium and also found in other organs such as the pituitary gland, encodes a G protein alpha subunit that mediates signal transduction by coupling with dopamine type 1 receptors and adenosine A2 receptors. This signaling plays critical roles in the olfactory epithelium, basal ganglia, and certain hormonal pathways [45]. In the pituitary gland, GNAL activates adenylyl cyclase and the cAMP signaling pathway, which mediates responses to mitogenic and secretagogue factors.

ITM2B is expressed in the human brain, pituitary gland, heart, pancreas, and liver, as reported in the Genotype-Tissue Expression (GTEx) RNA-seq dataset (accessible at https://www.gtexportal.org, URL accessed on 28 November 2024). ITM2B was initially identified as a modulator of the amyloid-beta A4 precursor protein (APP), inhibiting amyloid aggregation and fibril formation. In the pituitary gland, ITM2B promotes the stimulation and release of adrenocorticotropic hormone (ACTH), counteracting the inhibitory effects of guanylate cyclase [47].

Autoantibodies against GNAL and ITM2B have been linked to the development of hypophysitis. A subsequent cohort study confirmed these findings and demonstrated the presence of anti-GNAL autoantibodies in patients prior to immunotherapy who later developed hypophysitis, compared to those who did not. Some researchers have proposed a potential role for both autoantibodies as biomarkers of hypophysitis during treatment and have suggested GNAL autoantibodies as predictive biomarkers for the condition.

Hypophysitis associated with anti-CTLA4 antibodies and anti-PD1/PDL1 antibodies exhibits distinct clinical and pathological characteristics.

### 4.1. CTLA4-Ab-Related Hypophysitis

Beyond lymphoid cells, CTLA-4 has been identified on the surface of pituitary cells, particularly in prolactin- and thyrotropin-secreting cells. Pituitary autoantibodies targeting thyrotrophs, corticotrophs, and gonadotrophs have been detected in patients treated with anti-CTLA-4 antibodies (e.g., ipilimumab) who subsequently developed hypophysitis [11].

Experimental evidence from in vitro and murine models suggests that cytotoxicity in this context may be mediated by antibodies or complement activation. In CTLA-4 inhibitor-induced hypophysitis, a type II hypersensitivity reaction, followed by a type IV hypersensitivity reaction, has been implicated. These reactions involve complement fixation, phagocytosis, and the activation of autoreactive lymphocytes, triggering an inflammatory cascade. This inflammation does not always result in necrosis or fibrosis but consistently induces endocrine dysfunction.

The initial event is thought to involve the interaction between CTLA-4 antigens expressed on the pituitary cell surface and specific antibodies, leading to the formation of immune complexes and the recruitment of C1. This interaction activates the classical complement pathway, resulting in pituitary cell damage, macrophage infiltration, phagocytosis, and enhanced antigen presentation, which amplifies the immune response. Later events involve type IV hypersensitivity mechanisms, characterized by lymphocyte infiltration and the formation of ectopic lymphoid follicles near endocrine cells [11,49] (Figure 4).

Antibody-dependent cell-mediated cytotoxicity (ADCC) targeting T regulatory cells, triggered by CTLA-4 antibodies, has also been proposed as a mechanism. While this process was initially categorized under type II hypersensitivity, it is now classified as type VI hypersensitivity. However, further studies are required to fully clarify its role [50,51,52,53].

The IgG subclass of immune checkpoint inhibitors may influence the differential risk of hypophysitis. Monoclonal antibodies based on the IgG1 and IgG2 subclasses (e.g., ipilimumab and tremelimumab) are more likely to elicit ADCC compared to IgG4-based anti-PD1 monoclonal antibodies, as the former promote the activation of the classical complement pathway [52].

Variability in CTLA-4 expression among individuals, as indicated by the Genotype-Tissue Expression (GTEx) RNA-seq dataset (https://www.gtexportal.org, URL accessed on 28 November 2024), may partly explain the differences in the incidence and clinical presentation of hypophysitis following CTLA-4 antibody therapy [12].

In a case series of six autopsy patients, a correlation was documented between high levels of CTLA-4 expression on the surface membrane of pituitary cells and disease severity. This included findings such as B and T lymphocyte infiltration, occasional aggregation of ectopic lymphoid follicles, and emperipolesis [49].

Additionally, there is increasing evidence of an association between HLA haplotypes and the frequency of the disease. Notably, a Japanese study reported a significant link with HLA-Cw12 and HLA-DR15 [54,55].

### 4.2. PD1/PDL1-Related Hypophysitis

The pathogenesis of PD1/PDL1 hypophysitis remains unclear. The cell-mediated damage primarily affects corticotroph cells. It has been postulated that this is correlated with the expression of PD1 on the surface of pituitary cells (although the specific subset of these cells requires further clarification) and the expression of Fc receptors on corticotroph cells, which may render these cells more capable of binding anti-PD1 antibodies (Genotype-Tissue Expression RNA-seq dataset at https://www.gtexportal.org, URL accessed on 28 November 2024) [12].

Anti-PD1/PDL1 antibodies can induce the production of pituitary autoantibodies, including anti-corticotropin and anti-growth hormone (GH) antibodies, with suspected self-antigens such as growth hormone or opioid melanocortins. These antibodies are typically IgG4 or IgG1 with modified Fc regions. As a result, they are unable to activate the complement pathway and have reduced potency for eliciting antibody-dependent cellular cytotoxicity (ADCC). For these reasons, the pathogenesis of PD1/PDL1 hypophysitis is unlikely to align with type II or VI hypersensitivity but is instead considered to involve type IV hypersensitivity. Some researchers have also proposed a mechanism resembling primary IgG4-related hypophysitis [56,57].

Additionally, specific HLA haplotypes, including HLA-DQB1*06:01, HLA-DPB1*09:01, and HLA-DRB5*01:02, have been significantly associated with anti-PD1 antibody-related hypophysitis [55]. However, the mechanisms underlying these genetic associations remain to be elucidated.

### 4.3. New Item About IH: Paraneoplastic-Related Hypophysitis and Hypothalamitis

In a study analyzing 20 patients with PD-1/PDL-1-related hypophysitis who experienced isolated ACTH deficiency, circulating anti-proopiomelanocortin (POMC) antibodies were detected in approximately 10% of cases. Furthermore, analysis of their tumor tissues revealed ectopic expression of POMC protein. This finding led to the hypothesis that a subset of PD-1/PDL-1-related hypophysitis represents a latent form of paraneoplastic syndrome, a mechanism potentially shared with anti-pituitary hypophysitis and isolated ACTH deficiency. It has been proposed that ectopic expression of POMC or ACTH could evoke autoreactive T-cell activation, resulting in damage to pituitary corticotroph cells. Alternatively, this process may remain silent and latent, with immunotherapy acting as a trigger to enhance autoimmunity and cause specific injury to ACTH-secreting pituitary cells [15,58,59].

Recently, isolated hypothalamic autoimmunity has also been documented in a subset of patients undergoing immunotherapy [60]. An immune-mediated attack on hypothalamic cells that produce releasing factors has been hypothesized, possibly facilitated by PD-L1 expression on the surface of hypothalamic cells. This hypothesis is supported by findings from a prior study and a case report, though these assumptions require further investigation for confirmation [60,61,62] (Figure 5).

## 5. Clinical Presentation and Biochemical Diagnosis

The clinical presentation of IH is rather nonspecific. Detailed history and clinical examination are fundamental, notably for signs of underlying etiology with systemic symptoms [63]. For this, in advanced cancer patients, the diagnosis is complicated by the co-presence of systemic manifestations related to the neoplasm itself, possible brain/head/neck radiation and/or opportunistic infection. Systemic symptoms may include asthenia, anorexia, headache, vomiting, weight loss, low blood pressure, dizziness, decreased libido, hot flashes, and, more rarely, visual disturbances [55]. Even rarer are polyuria and polydipsia [56].

First of all, when approaching the topic of hypophysitis, it is necessary to clarify some distinguishing points between IH and primary lymphocytic hypophysitis.

Compared to the more common form of primary lymphocytic hypophysitis, IH affects mostly men than women, especially in the sixth decade of life, with a prevalence of anterohypophysial axes deficits. The involvement of neurohypophysis is rarer, and if present, requires further investigations to exclude any metastasis from primary neoplasm. Other autoimmune endocrine diseases usually follow the onset of IH, unlike what happens in the more common lymphocytic form [63,64,65].

In a multicenter retrospective cohort study, which included 56 patients with IH and 69 patients with primary hypophysitis, the greater susceptibility of the male sex in the non-primary form was confirmed. They showed a greater involvement of more anterohypophysial axes. The thyrotropic and gonadotropic were in addition to the most frequent corticotropic common in both sexes. Visual field involvement, as well as diabetes insipidus, occurred less frequently in IH. A total of 20% of the patients in this subgroup did not report clinical symptoms at diagnosis [64].

It should be noted that the diagnostic criteria for immune checkpoint inhibitor-induced hypophysitis (IH) are not yet standardized. Some authors propose that a diagnosis of IH can be made when there are ≥2 pituitary hormone deficits (including secondary hypothyroidism or hypoadrenalism) or ≥1 pituitary hormone deficiency accompanied by magnetic resonance imaging (MRI) abnormalities in the presence of suggestive symptoms [66].

The occurrence of hyponatremia, hypotension, or hypoglycemia in patients undergoing immunotherapy should raise suspicion of IH and prompt further endocrinological evaluation [67]. IH onset occurs approximately 9 weeks (with a range of 5–36 weeks) after the beginning of the therapy [68,69]. In patients receiving low-dose ipilimumab, the median time to onset is delayed (11 weeks) compared to those receiving high-dose therapy, suggesting a potential cumulative effect of repeated drug doses [70,71].

Overall, anterior hypopituitarism is more common than diabetes insipidus [67,72,73,74]. The most frequent hormonal deficiencies involve ACTH and/or thyroid-stimulating hormone (TSH), while prolactin levels may be either elevated or reduced [75,76]. Additional findings include hypogonadotropic hypogonadism and low levels of insulin-like growth factor-1 (IGF-1).

As for the pathophysiology described above, also for clinical presentation there are differences between anti-CTLA-4-induced hypophysitis and anti-PD-1/PD-L1-induced hypophysitis [Table 2].

Overall, El Osta et al. analyzed these differences and concluded that the incidence of any grade immune-related adverse event was higher in patients who received ICI targeting CTLA-4 (53.8%) than PD-1 (26.5%) and PD-L1 ICI (17.1%) (*p* < 0.001) [77].

In a large meta-analysis of 38 clinical trials including 7551 patients, the incidence of endocrine dysfunction was significantly higher in those treated with combination therapy compared with ipilimumab [78]. Regarding monotherapy, the incidence of thyroid dysfunction and IH was highest with programmed cell death protein 1 inhibitors and with ipilimumab, respectively [78].

Further meta-analyses have highlighted how the risk of immune-related adverse events was higher with combination therapies [79,80] and how the incidence and severity of these events were often drug- and dose-independent [81].

Specifically, about IH, there are two distinct patterns of presentation: a lymphocytic hypophysitis-like condition with pituitary enlargement and multiple hormone deficiencies in those treated with anti-CTLA-4 agents/combination CTLA-4/PD-1 therapy; isolated ACTH deficiency in those treated with anti- PD-1 agents [82].

The incidence of IH after ipilimumab (3 mg/kg) treatment varies from 3% to 17%, depending on the study and adverse event definition [83]. However, a greater incidence of 25% has been documented for patients on a dosage of 10 mg/kg [84].

The risk of hypophysitis described in published reports may be inaccurate due to variations in diagnostic approaches, delays in diagnosis due to the concurrent use of glucocorticoids and chemotherapy for other immune-related adverse events, and a lack of targeted pituitary imaging at symptom onset [83].

IH commonly appears 2 to 3 months after starting ipilimumab treatment or after the third dose (given every 3 weeks) [84,85]; however, cases of later development have also been reported [72]. Some studies showed that ipilimumab regimens with dosages of 10 mg/kg are more likely to develop hypophysitis than regimens with doses of 3 mg/kg [83,84], but recent studies did not support this [86]

Clinically, ipilimumab-induced hypophysitis is often associated to headache and fatigue; other symptoms—already described above—can be confused with more generic symptoms related to the tumor itself or even to intercurrent therapies (e.g., nausea, anorexia).

The most frequent hormonal deficiencies in these patients are secondary adrenal insufficiency and hypothyroidism, a consequence of the cellular trophism of ipilimumab and the cause of the main symptoms of the patients [83]. Patients with severe ipilimumab-induced hypophysitis may rarely experience visual abnormalities due to pituitary gland enlargement, although hyponatremia and/or cardiovascular compromise due to severe secondary adrenal insufficiency are more common at presentation [83].

Other pituitary axes can also be affected in ipilimumab-induced hypophysitis, resulting in secondary hypogonadism, hypoprolactinemia, and growth hormone (GH) insufficiency [85]. However, these endocrinopathies are less clinically significant during the acute phase.

Pituitary MRI in selected patients with ipilimumab-induced hypophysitis revealed radiologic abnormalities such as widespread enlargement of the gland and varied enlargement of the stalk [83,84,85]. Pituitary MRI may help diagnose hypophysitis and, last but not least, exclude differential diagnoses.

Anti PD-1 or anti PD-L1 monotherapy causes hypophysitis less frequently than anti-CTLA-4 therapies.

These drugs have a greater impact on the corticotropic axis and cause secondary adrenal insufficiency, often due to isolated ACTH deficiency [82].

Clinically, a smaller proportion of patients present with headache and have pituitary enlargement on MRI [83]. Multiple studies have linked secondary adrenal insufficiency to nivolumab-induced thyroid impairment [87,88].

In a retrospective, multicentre study, patients treated with nivolumab or pembrolizumab had a 0.5% prevalence of hypophysitis, with symptoms onset later than those treated with ipilimumab (a median of 26 weeks of treatment, wherein PD-1 inhibitors are typically dosed every 2–6 weeks, depending on the agent and dose). All hypophysitis cases were diagnosed with secondary adrenal insufficiency, with rare cases of hypogonadism, GH deficiency, or hypoprolactinemia [89].

So, it can be stated that hypophysitis secondary to nivolumab and pembrolizumab is a clinical entity distinct from ipilimumab-induced hypophysitis [65].

Finally, some authors recently investigated the publicly available US FDA Adverse Event Reporting System (FAERS) database to gain insight into the possible association between immune checkpoint inhibitors and hypophysitis [90]. In this article, ipilimumab, nivolumab, pembrolizumab, and atezolizumab were, in this descending order, statistically correlated with the target adverse event [90], confirming what was previously described.

**Table 2 cancers-17-00436-t002:** Differences between anti-CTLA-4/anti-PD-1/anti PD-L1-induced hypophysitis.

	Anti-CTLA-4-Induced Hypophysitis	Anti-PD-1/PD-L1-Induced Hypophysitis
**Patterns of presentation**	Lymphocytic hypophysitis-like condition with multiple hormone deficit [82]	Isolated ACTH deficiency [82]
**Common onset time**	2 to 3 months after starting treatment or after the third dose [84,85]	A median of 26 weeks of treatment [89]
**Prevalent hormonal deficits**	Hypothyroidism and secondary adrenal insufficiency [83]	Secondary adrenal insufficiency [82]
**Local mass effects**	Can be present [84,85,86]	Less frequent [85]
**Enlargement of pituitary at MRI**	Approximately 98% of cases [83]	Approximately 28% of cases [83]

Routine biochemical assessments, including electrolyte measurements, thyroid function adrenal function, and gonadal function, can highlight biochemical/hormonal alterations suggestive of IH. In this case, it is appropriate to perform MRI of the pituitary with contrast medium not only to diagnose IH but also to exclude other diagnoses (e.g., metastases) [91].

When is biochemical monitoring appropriate during immunotherapy?

At baseline;At each cycle of therapy for the first 6 months;Every two cycles of therapy for the following 6 months;Subsequently in the presence of clinical suspicion.

This timing is suggested on the basis of the IH onset reported in the literature, but it can change depending on individual clinical findings [92].

Specifically at baseline, it is recommended to measure fasting blood sugar (only with anti-PD-1/PD-L1), plasma sodium, TSH, FT4, 08:00 h cortisol (in the absence of exogenous corticosteroids), LH, FSH, testosterone in males, and FSH in females post-menopause; ACTH measurement should be performed in patients with 08:00 h cortisol <500 nmol/L. LH, FSH, and estradiol should be measured in premenopausal women with irregular periods after the exclusion of other non-iatrogenic etiologies [92].

Once immunotherapy is started, the following tests are recommended at the above time points: fasting blood sugar (only with anti-PD-1/PD-L1), plasma sodium, TSH, FT4, cortisol, and testosterone in males [92].

## 6. Radiological Findings

In the presence of suspicious symptoms of IH, MRI is mandatory. MRI can confirm IH and exclude other causes of pituitary deficiencies, such as pituitary metastases, apoplexy, and abscesses [93,94,95]. In the context of differential diagnoses, it should be noted that other space-occupying lesions of the brain can also cause pituitary hormonal deficits similar to those related to IH.

Computed tomography (CT) can also assess gland size, but other features of IH and potential differential diagnoses are not as readily assessable. When readily available, integrated 2-[18F]-fluoro-2-deoxy-D-glucose positron-emission tomography (PET)/computed tomography (CT) can provide complementary diagnostic aspects, but it is not primarily used for the assessment of suspected IH and MRI remains the imaging technique of first choice [96].

In the case of IH, MRI shows a modest pituitary enlargement (often < 2 cm), stalk thickening, and allogeneic or heterologous contrast enhancement in 77% of cases, which is less common than that in primary hypophysitis [55]. These findings are particularly related to anti-CTLA-4 antibodies rather than anti-PD-1/anti-PD-L1 antibodies [97,98]. As mentioned above, however, the diagnosis of IH can also be made in the absence of concomitant specific radiological signs; in this regard, it should be considered that sometimes the pituitary anatomical alterations can precede the biochemical alterations, and imaging is therefore often carried out when this alteration has already regressed [56,69,86,99]. In IH, the gland size returns to baseline size or smaller within months. Larger pituitary size, the presence of a discrete lesion surrounded by normal pituitary tissue, deviation of the pituitary stalk, sella expansion, and clival invasion are not typical of IH and suggest other diagnoses [96]. Finally, in these oncologic patients, careful observation of symptoms and multidisciplinary assessment are required.

From the precise perspective of a multidisciplinary management of IH, in the diagnostic phase, it is sometimes possible to obtain a neurosurgical consultation for the differential diagnosis. However, a minority of patients will undergo a diagnostic surgical procedure, as careful clinical evaluation may orient toward a probable diagnosis [63].

Finally, once the diagnostic orientation is for IH, the therapy does not include surgical treatment but a medical approach, as described below.

## 7. Therapeutic Approach

Early diagnosis and appropriate treatment of IH are essential as it can manifest as a life-threatening condition due to an adrenal crisis [55].

Patients experiencing acute unwellness or with signs of adrenal crisis should be managed according standard protocoll, as outlined in the Society for Endocrinology guidelines for the acute management of the endocrine complications of immunotherapy [100]. These guidelines show three algorithms to manage endocrine toxicities via immunotherapy on the basis of their gravity, specifically, endocrine management of patients in the first 24 h who present as life-threateningly unwell (CTCAE grade 3–4) and the appropriate management of mild–moderately unwell patients (CTCAE grade 1–2) presenting with features compatible with endocrinopathy.

Regarding IH, in the case of severe, potentially life-threatening conditions or possibly adrenal insufficiency, urgent management is required:Hydrocortisone (immediate bolus injection of 100 mg hydrocortisone i.v. or i.m., followed by continuous intravenous infusion of 200 mg hydrocortisone per 24 h (alternatively, 50 mg hydrocortisone per i.v.or i.m. injection every 6 h)).Rehydration with rapid intravenous infusion of 1000 mL of isotonic saline infusion within the first hour, followed by further intravenous rehydration as required (usually 4–6 L in 24 h; monitor for fluid overload in case of renal impairment and in elderly patients) [101].

Once the clinical and biochemical situation has been stabilized:Convert to oral hydrocortisone (initially 20 mg in the morning/10 mg in the afternoon/10 mg in the evening mg to reduce to maintenance of 10 mg in the morning/5 mg in the afternoon/5 mg in the evening) or oral prednisolone (maintenance 3–5 mg per day).Consider primary adrenal failure (because immunotherapy can also cause adrenalitis): assess renin/aldosterone (particularly if ACTH elevated/normal and hyponatremia present).Continue immunotherapy if no other contraindications [100].

In the case of patients with mild/moderate symptoms, a biochemical endocrinological screening is required. In the case of adrenal insufficiency, it is mandatory to start oral hydrocortisone. In an acute setting, in the case of borderline value, ACTH (1–24) stimulation testing may not accurately diagnose acute secondary adrenal insufficiency due to the adrenal glands’ ability to respond to exogenous ACTH stimulation after many weeks of atrophy [83].

All patients with adrenal insufficiency should be provided with a steroid emergency card, a hydrocortisone emergency injection kit, and education with regard to “sick day rules” [100,101,102,103].

Usually, adrenal insufficiency related to IH is permanent [55,104,105,106].

Regarding steroid therapy, it should be noted that beyond endocrine replacement therapy, high-dose glucocorticoids are not recommended except for serious symptoms, such as severe headache, diplopia, or visual deficit due to severe pituitary enlargement, in order to reduce the mass effect [107]. If these mass effect symptoms are present, methylprednisolone 1–2 mg/kg daily (or its equivalent) for 3–5 days can be started, followed by oral prednisone 1–2 mg/kg, with gradual reduction in 4 weeks [52]. If methylprednisolone or other pharmacological dose glucocorticoids are administered for these concomitant events, additional hydrocortisone is not required until high-dose steroid therapy is discontinued. Once the acute phase is over, the patient who has developed adrenal insufficiency will be treated with the previously reported steroid replacement dose therapy.

The efficacy of supraphysiologic glucocorticoids in these cases has not been evaluated prospectively [83]. However, retrospective studies have shown short-term symptom resolution without a control group exposed to lower doses [83]. Patients with simultaneous immunotherapy-related adverse events may require substantial doses of glucocorticoids to treat, in addition to IH [83].

A retrospective analysis found that individuals with melanoma and hypophysitis who received high-dose glucocorticoid regimens had lower survival rates compared to those who received ≤7.5 mg of prednisone equivalents daily [85]. Other studies found that large doses of glucocorticoids did not increase pituitary function recovery [69,84]. This raises the possibility that short-term stress dosing during acute illness, followed by physiologic or less aggressive glucocorticoid replacement regimes, could be equally helpful [83].

Central hypothyroidism and hypogonadism are the other frequent deficits related to IH and should be treated accordingly, even if their correction does not require an urgent approach as for hypocortisolism [100,105]. In the case of coexistence of hypocortisolism and hypothyroidism, hypocortisolism must be corrected before starting replacement therapy with levothyroxine for hypothyroidism.

Furthermore, unlike adrenal insufficiency, both hypothyroidism and hypogonadism related to IH can be transient [69,72,84,86,108,109]. For this reason, a constant monitoring of biochemical values is recommended in order to highlight the possible recovery of the functions; this recovery can be assessed if, over time, it is possible to suspend the replacement therapies in question.

Treatment of GH deficiency related to IH is not recommended because the therapy with recombinant GH is inappropriate in patients with active cancer [110,111].

The need for cancer theraphy suggests that the onset of IH does not contraindicate the continuation of immunotherapy, which can possibly be postponed based on the patient’s clinical conditions [56,69,112,113].

Finally, an important emerging aspect is that, based on current knowledge on the endocrine adverse effects related to immunotherapy, one can consider outpatient management of patients with IH.

In this regard, in a recent study, authors analyzed ambulatory emergency care for patients with IH [114].

In this study, in the absence of severe symptoms (sodium <125 mmol/L, hypotension, reduced consciousness, hypoglycemia, and/or visual field defect), patients were given a single intravenous dose of hydrocortisone (100 mg), observed for at least 4 h, and discharged on oral hydrocortisone (20 mg, 10 mg, and 10 mg) [114]. Patients were then seen quickly in an endocrinologist outpatient setting for additional treatment. The pathway was used to manage 14 patients with biochemically proven ACTH insufficiency. As many as 7 of the 14 patients had combination ICI therapy, whereas 4 had pan-anterior hypopituitarism. There were no 30-day readmissions or associated hypophysitis-related deaths, and all patients continued their ICI therapy without interruption [114].

This type of management highlights the importance of multidisciplinary teams and dedicated pathways in evaluating these adverse events of immunotherapy in the near future.

## 8. Conclusions

Hypophysitis is an important endocrine adverse event of immunotherapy. Its presentation is often a diagnostic challenge, even for the most experienced clinicians. As more patients continue to be placed on immune checkpoint inhibitors for the treatment of cancers, there is a growing need for a multidisciplinary approach to managing their complications. Appropriate diagnosis and management are important because this condition, if not recognized, can be fatal for the patient.

## Figures and Tables

**Figure 1 cancers-17-00436-f001:**
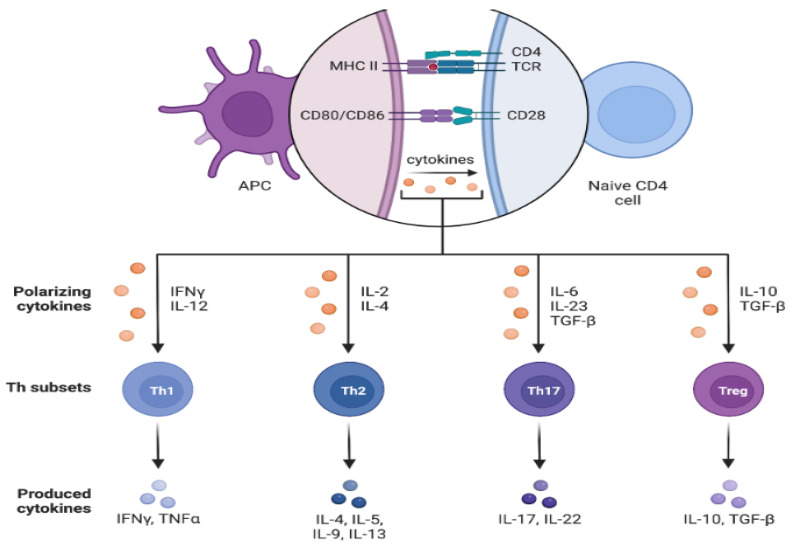
T-cell activation and differentiation evoked by the binding of TCR to MCH II, by CD28 to CD80/86, and by the concurrent binding of polarizing cytokines to their respective receptors on the T-cell surface. Created in BioRender. La Sala, D. (2025) https://BioRender.com/l40l288. Abbreviations: MCH II: major histocompatibility complex type II; CD4: cluster of differentiation 4; TCR: T cell- receptor; CD80/86: cluster of differentiation 80/86; CD28: cluster of differentiation 28; APC: antigen presenting cell; IFNγ: interferon-γ; IL-12: interleukin 12; IL-2: interleukin 2; IL-4: interleukin 4; IL-6: interleukin 6; IL-23: interleukin 23; TGF-β: transforming growth factor-β; IL-10: interleukin 10; Th1: T helper 1; Th2: T helper 2; Th17: T helper 17; T reg: regulatory T cells; TNFα: Tumor necrosis factor α; IL-4: interleukin 4; IL-5: interleukin 5; IL-9: interleukin 9; IL-13: interleukin 13; IL-17: interleukin 17; IL-22: interleukin 22; IL-10: interleukin 10.

**Figure 2 cancers-17-00436-f002:**
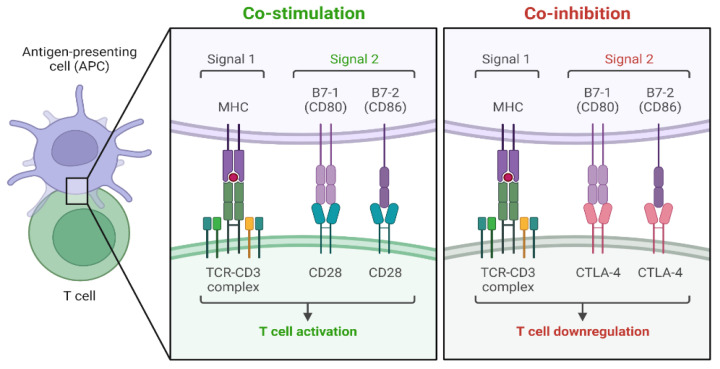
The mechanisms by which the TCR activation signal is increased or suppressed by costimulatory or coinhibitory signals. Created in BioRender. La Sala, D. (2025) https://BioRender.com/d25o366. Abbreviations: MHC: major histocompatibility complex; CD80: cluster of differentiation 80; CD86: cluster of differentiation 86; TCR-CD3 complex: T cell receptor- cluster of differentiation 3 complex; CD28: cluster off differentiation 28; CTLA-4: cytotoxic T-lymphocyte–associated antigen 4.

**Figure 3 cancers-17-00436-f003:**
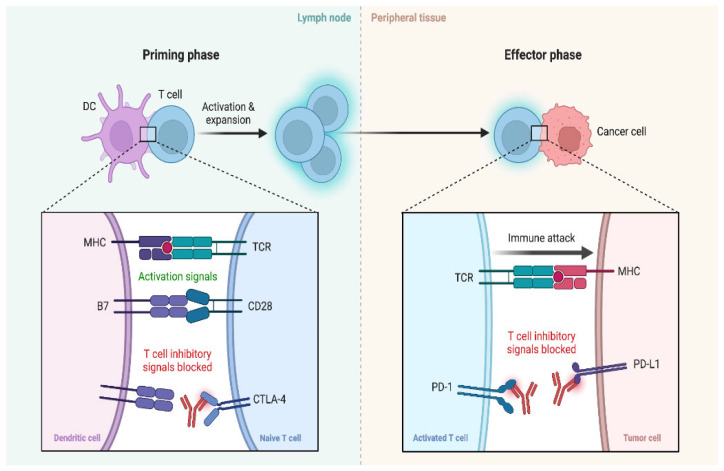
Awakening of the immune response against tumor cells by monoclonal antibodies against CTLA4, PD1, or PDL1. Created in BioRender. La Sala, D. (2025) https://BioRender.com/t81f332. Abbreviations: DC: dendritic cell; MHC: major histocompatibility complex; TCR: T cell receptor; CD28: cluster of Differentiation 28; CTLA-4: cytotoxic T-lymphocyte–associated antigen 4; PD-1: programmed cell death protein 1; PD-L1: programmed cell death ligand 1.

**Figure 4 cancers-17-00436-f004:**
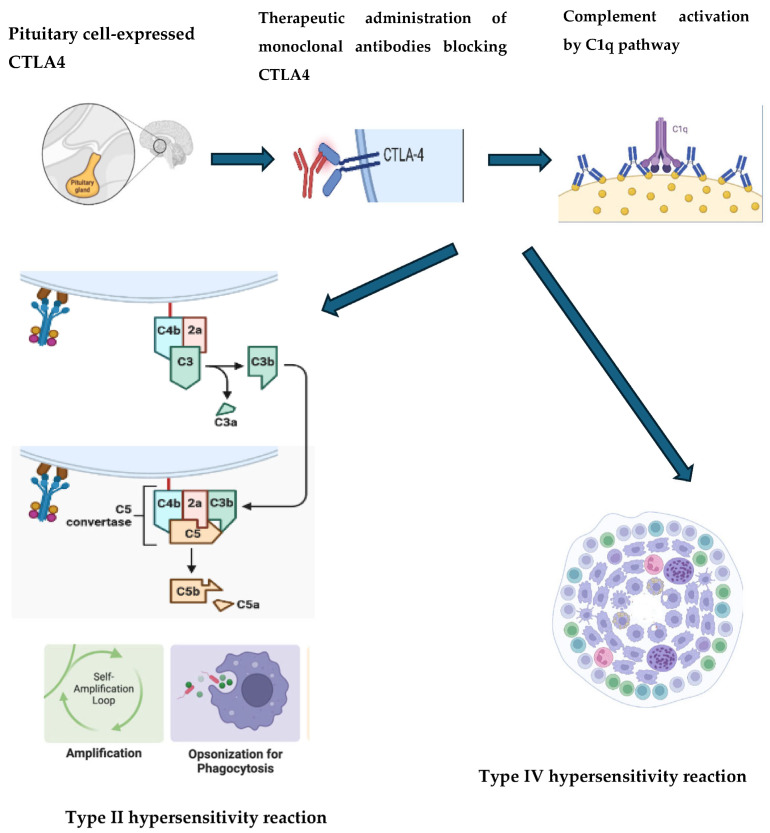
Representation of the mechanism of action of CTLA-4 blocking antibodies on pituitary cells. The immune complex between antibodies and antigens recruits complement C1q molecules that active the classical complement pathway, with subsequent pituitary damage and the recruitment of macrophages and other inflammatory cells leading to phagocytosis and enhanced antigen presentation (type II hypersensitivity reaction). The lymphocytes infiltration with ectopic lymphoid follicles near the pituitary cells (type IV hypersensitivity reaction) are, instead, a late event. Created in BioRender. La Sala, D. (2025) https://BioRender.com/d32a44. Abbreviations: CTLA-4: cytotoxic T-lymphocyte–associated antigen 4.

**Figure 5 cancers-17-00436-f005:**
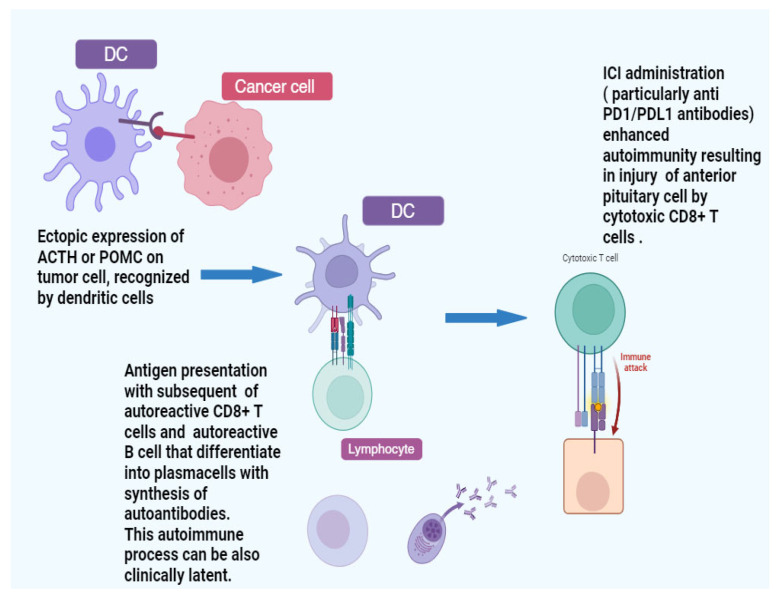
Postulated mechanism of action of PD1/PDL1 blocking antibodies on pituitary cells. It was supposed that at least a part of PD1/PDL1 antibodies-related hypophysitis is an enhancement of an already-present endogenous paraneoplastic form, evoked by an ectopic ACTH or POMC expression by tumor cells. Created in BioRender. La Sala, D. (2025) https://BioRender.com/r57p193. Abbreviations: DC: dendritic cell; ACTH: adrenocorticotropic hormone; POMC: pro-opiomelanocortin; CD8: cluster of differentiation 8; ICI: immune checkpoint inhibitor; PD1: programmed cell death protein 1; PDL1: programmed cell death ligand 1.

**Table 1 cancers-17-00436-t001:** Immune checkpoint inhibitors and their clinical indications.

Drug	Target	Location	Indication	References
Ipilimumab	CTLA-4	T-cell	Melanoma, RCC, colorectal cancer	[27]
Nivolumab	PD-1	T-cell	NSCLC, melanoma, renal cell cancer, melanoma, NSCLC, SCLC, RCC, Hodgkin lymphoma, SCC of H&N, urothelial carcinoma, colorectal carcinoma, HCC	[28,29]
Pembrolizumab	PD-1	T-cell	Metastatic melanoma, NSCLC, Hodgkin lymphoma, SCC of H&N, urothelial carcinoma, gastric tumors, bladder cancer, head and neck cancer, esophageal cancer, cervical cancer, HCC, RCC, Merkel cell carcinoma, triple negative breast cancer, colorectal carcinoma	[2,30,31,32,33,34]
Atezolizumab	PD-L1	Tumor cell	NSCLC, urothelial carcinoma, SCLC, triple negative breast cancer, unresectable HCC	[35]
Durvalumab	PD-L1	Tumor cell	Urothelial carcinoma, unresectable locally advanced NSCLC	[36]
Avelumab	PD-L1	Tumor cell	Merkel cell carcinoma, urothelial carcinoma, RCC	[37,38,39]
Cemiplimab	PD-1	T-cell	Cutaneous SCC	[40]
Relatlimab + Nivolumab	LAG-3 + PD-1	T-cell	Unresectable or metastatic melanoma	[26,41]

## Data Availability

Not applicable.

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
