# Peer review of "Immunotherapy-Related Hypophysitis: A Narrative Review"

_cancers, 2025, doi:10.3390/cancers17030436_

Round 1

Reviewer 1 Report (New Reviewer)

Comments and Suggestions for Authors

The manuscript "Immunotherapy-related hypophysitis: a narrative review" by Di Stasi et al. is a comprehensive review focusing on the immunological aspects of immunotherapy-related hypophysitis. The manuscript obviously presents a revised version. The revised version incorporates additional clinical content, broadening its scope and relevance. However, this revision introduces some issues that merit attention and refinement.

The inclusion of clinical aspects enriches the manuscript, but further elaboration is necessary for a comprehensive review. For example:

  - There is no detailed comparison between the clinical presentation of ICI-induced hypophysitis and primary hypophysitis. This distinction is clinically significant and should be discussed (e.g., see Amereller 2022, DOI: 10.1007/s11102-021-01182-z; Atkins & Ur 2020, DOI: 10.1080/07435800.2020.1817064).

  - Differences in clinical manifestations associated with specific ICI agents, as outlined by Faje (2019) and Langlois (2021, DOI: 10.1210/clinem/dgab672), should be explored in greater depth. Relevant literature such as Barroso-Sousa (2018, DOI: 10.1001/jamaoncol.2017.3064), El Osta (2017, DOI: 10.1016/j.critrevonc.2017.09.002), de Filette (2019, DOI: 10.1055/a-0843-3366), Almutairi (2020, DOI: 10.3389/fonc.2020.00091), Da (2020, DOI: 10.3389/fphar.2019.01671), and van der Leij (2024, DOI: 10.3389/fendo.2024.1400841) should be looked into.

  - The recently published article by Tang et al. (2024, DOI: 10.1097/MD.0000000000037587) may also provide relevant insights worth incorporating. (Admittedly, it's not an outstanding piece of literature, but interesting nonetheless.)

Minor Comments:

- The manuscript would benefit from an additional round of editing to enhance the flow and coherence of the added sections.

- Harmonizing the tone and style of writing across all sections would improve the overall quality and readability.

- The figures included in the manuscript are of varying quality. Text labels embedded in the text document (e.g., page 8) should be integrated into the corresponding figures for better clarity and presentation (as is the case for fig. 5 on page 10). Since no sources are cited for the figures, it is assumed they are original. If so, it should be relatively straightforward to standardize their appearance.

- The phrase "Equally contributed to study" is misapplied in this context, as this is a review, not a study. Consider rephrasing to "Equally contributed to the manuscript" or similar.

- For clarity and international recognition, consider adding the term "ACTH (1-24)" alongside "cosyntropin," as the latter is not universally familiar.

In summary, this manuscript addresses an important topic with significant clinical implications. While the revisions are a step in the right direction, they require further refinement and expansion.

Comments on the Quality of English Language

While the manuscript is generally well-written and fluid, the newly added clinical sections are less cohesive and read as fragmented thoughts. To improve readability and coherence, these sections would benefit from reorganization and integration into the main narrative.

Author Response

Please see the attachment that we've uploaded. 

Reviewer 2 Report (New Reviewer)

Comments and Suggestions for Authors

The authors provide a narrative review on pituitary/sellar inflammation (hypophysitis) as a complication of immune check point cancer therapy (ICI).  The manuscript is reasonable but some areas propagate simplistic traditional views which need to be updated.

1. the term "pituitary swelling" needs to removed throughout as it reflects a poor understanding of the pathobiology of the spectrum of hypophysitis. Indeed, the radiographic-clinical correlation can be discordant with some patients showing no radiographic evidence of sellar enlargement with many showing an "empty sella".  Further, the compromise of parasellar structures (infidibulum and optic chiasm) is more often due to infiltration as opposed to neoplastic compression by a "swollen pituitary". 

2. Loss of anterior and/or posterior pituitary function can result from non- ICI conditions in cancer patients including brain/head/neck radiation and/or opportunistic infections (eg: COVID-19 etc.). These considerations need to be included in the differential diagnoses.

3. The indications for neurosurgical consideration for diagnostic and/or therapeutic intentions in this clinical setting need to be articulated. 

4. Synthetic glucocorticoids, such as prednisone, are not the choice for cortisol replacement.  The latter is better achieved with oral hydrocortisone.  Further, as the authors point out, the evidence supporting the therapeutic benefits of prednisone in ICI-induced IH remains to be shown.  

5. MRI examples of patients with documented IH, ideally with serial imaging, should be shown as illustrative examples.

6.  A table summarizing pituitary regulated hormones in males/females/menopausal groups should be shown in the diagnostic section. 

Comments on the Quality of English Language

The manuscript would benefit from complete editing by an English-speaking writer. 

Author Response

Please see the attachment that we uploaded.

Round 2

Reviewer 2 Report (New Reviewer)

Comments and Suggestions for Authors

The authors have responded to several of the questions and comments raised earlier. 

This manuscript is a resubmission of an earlier submission. The following is a list of the peer review reports and author responses from that submission.

Round 1

Reviewer 1 Report

Comments and Suggestions for Authors

Summary: The authors provide a brief overview of immune-check point inhibitor induced hypophysitis. Unfortunately, significant editing is required to make the paper easier to read. Furthermore, details provided regarding the epidemiology, diagnosis and management are brief and simplistic compared to the description of the immune system. While some pathophysiology is highlighted, how certain antibodies outlined in the manuscript and their use in clinical practice were not discussed.

Comment:

On page 1- where the heading states “Simple Summary:” a simple summary was not provided

I suspect the intent of the “Overview of basic principles of T cell immune response to antigen” was to simply and effectively convey the immunomodulatory aspect of the immune system and how ICI could adversely impact this leading to IR-AE , however I do not believe this occurred. The explanation is complex with additional information provided that is not critical. For example line 75 “It consists in an extracellular domain, a transmembrane domain and a cytoplasmatic tail, and is encoded by a 4-exon gene located on chromosome 2 in humans” is not necessary with regards to what the title suggests the intent of the manuscript is.

Edits: significant editing of the English language is required throughout this document to make it comprehensible to the reader. Only are few are highlighted here as examples

Page 3 line 114 “The seconds, instead, usually haven’t cytotoxic or phagocytic properties, but are in- volved in the recruitment of other types of cellular and humoral immunity. It’s possible to distinguish between CD4+ cells a subgroup, the regulatory T cells (TregCD4+CD25+), which have the CTLA4 antigen constitutively expressed, with an immunosuppressive function.” 

This should read:

The latter group (CD4+ T cells), instead, do not have cytotoxic or phagocytic properties, but are involved in the recruitment of other types of cellular and humoral immunity. It is possible to distinguish between CD4+ cell subgroups, regulatory T cells and effector T cells. Regulatory T cells (TregCD4+CD25+) constitutively expresses CTLA4 and have an immunosuppressive function……….

Line 147 “The expression of CD80/86 by tumor cells causes, in the tumor infiltrating lymphocytes, the translocation of CTLA4 from the cytoplasm to the cell surface of which is thus overexpressed, determining the suppression of the activity of the T lymphocytes themselves”

Should be changed to

“The expression of CD80/86 by tumor cells causes the translocation of CTLA4 to the cell surface from the cytoplasm in the tumor infiltrating lymphocytes resulting in suppression of T lymphocyte activity.”

Please provide a reference for this “As occurs in other autoimmune conditions, a subset of Th17 cells can switch to Th1 phenotype, retaining some of Th17 features and displaying a more aggressive behavior”

The authors propose a monitoring schedule for IH on page 11 including

• At baseline

 • At each cycle of therapy for the first 6 months

• Every two cycles of therapy for the following 6 months

 • Subsequently in the presence of clinical suspicion.

However, no reference is provided to support this recommendation, and this is stipulated in the guidelines (ASCO or AACE). If this recommendation is one based on the authors experience, then this should be clearly stated so as not to mislead the reader. Furthermore, what testing do the authors suggest?

The authors suggest that “MRI can confirm IH and exclude other causes of pituitary deficiencies such as pituitary metastases…..” unfortunately this is not always the case and although there are some radiological features that can help identify hypophysitis from other sellar pathologies this is not perfect and therefore close follow up is required.

The authors recommend in each scenario that an MRI be performed, however they mention that even in the absence of radiological findings a diagnosis can be made. This therefore what is the utility of performing an MRI?  I would suggest that the authors recommend an MRI to see if there is findings suggestive of hypophysitis but also to exclude other pathologies which could lead to pituitary deficiencies in the absence of radiological findings consistent with hypophysitis.

Line 409 . “In case of adrenal insufficiency, it is mandatory to start oral hydrocortisone while borderline values need further investigations such as Synachten stimulation test and close biochemical monitoring including electrolytes” Why do the authors suggest that close monitoring of electrolytes is required? This is not used to routinely diagnose or monitor for secondary adrenal insufficiency, so its value is not clear. Additionally, I would like to draw attention to the fact that in acute settings an ACTH stimulation test (Synacthen test) could be normal and therefore may exclude adrenal insufficiency.

Page 13 line 417 “Regarding steroid therapy, it should be noted that, beyond endocrine replacement 417 therapy, high-dose glucocorticoids are not recommended except for serious symptoms 418 such as severe headache, diplopia, or visual deficit due to severe pituitary swelling [97].” The authors do not highlight why this is the case. This is contrary to autoimmune hypophysitis and is therefore important for the readers to know but for the as this could impact outcome. Furthermore, it is important to outline the controversies surrounding this concept.

On page 13 line 433 “For this reason, it is recommended a constant monitoring of biochemical values in order 433 to highlight a possible recovery of the functions.” The author indicate that constant monitoring is recommended for recovery however this is not likely to indicate recovery. The patients will need to be weaned from the medication and reassessed to determine recovery.

Comments on the Quality of English Language

please see comments to the author regarding the English language editing required in short 

Unfortunately, significant editing is required to make the paper easier to read. 

Edits: significant editing of the English language is required throughout this document to make it comprehensible to the reader. Only are few are highlighted here as examples

Page 3 line 114 “The seconds, instead, usually haven’t cytotoxic or phagocytic properties, but are in- volved in the recruitment of other types of cellular and humoral immunity. It’s possible to distinguish between CD4+ cells a subgroup, the regulatory T cells (TregCD4+CD25+), which have the CTLA4 antigen constitutively expressed, with an immunosuppressive function.” 

This should read:

The latter group (CD4+ T cells), instead, do not have cytotoxic or phagocytic properties, but are involved in the recruitment of other types of cellular and humoral immunity. It is possible to distinguish between CD4+ cell subgroups, regulatory T cells and effector T cells. Regulatory T cells (TregCD4+CD25+) constitutively expresses CTLA4 and have an immunosuppressive function……….

Line 147 “The expression of CD80/86 by tumor cells causes, in the tumor infiltrating lymphocytes, the translocation of CTLA4 from the cytoplasm to the cell surface of which is thus overexpressed, determining the suppression of the activity of the T lymphocytes themselves”

Should be changed to

“The expression of CD80/86 by tumor cells causes the translocation of CTLA4 to the cell surface from the cytoplasm in the tumor infiltrating lymphocytes resulting in suppression of T lymphocyte activity.”

Author Response

Dear Editors and Reviewers,
Thank you for your letter and for the reviewers’ comments. Those comments are all valuable and very helpful for revising and improving our paper. We have studied comments carefully and have made correction which we hope meet with approval. Answers to the reviewer’s comments are as flowing:

Answers to Reviewer 1

We have written the requested  simple   summary.

We have edited the quality of the english language and we have changed some expressions  and sentences as you  requested .   We checked the spelling mistakes and corrected them.

We have provided to insert the reference for the expression: “……As occurs in other autoimmune conditions, a subset of Th17 cells can switch to Th1 phenotype, retaining some of Th17 features and displaying a more aggressive behavior. This result was not confirmed by a similar study with human pituitary tissue of patients with hypophysitis (three of whom with IH), where only high transcript level of IL17-A was detected of but not of INFɤ …..” .

This bibliographic reference is the same as the previous one ( reference n. 52).

For the clinical paragraphs, we have emphasized more exhaustively the differences between CTLAT4/PD-1/PD-L1 induced hypophysitis. In this regard, a table has also been added (table 2). Figure 6 has been eliminated.  The clarifications requested by reviewer 1 have also been described. 

Reviewer 2 Report

Comments and Suggestions for Authors

The authors present a review of ICI mediated hypophysitis. There are a significant number of similar reviews already in the literature. This a further well written review but there are some issues which need to be addressed:-

1. In a narrative review of hypophysitis I find the review in sections 2 and 3 (whilst well written) entirely superfluous to the topic at hand.

2. I think the clinical presentation and diagnostic work-up section needs major revision. There are 2 well defined clinical presentations now clearly defined in the literature - CTLA-4/combination therapy giving rise to a lymphocytic-like hypophysitis and isolated ACTH deficiency in PD-1 and rarely PDL-1 therapy. This leads to a different diagnostic and imaging approach. I feel figure 6 is entirely wrong in that context. There are large numbers of papers/position statements etc.. outlining this. 

3. The other key point is that the vast majority of these patients can be now be managed in an emergency outpatient/ambulatory setting and do not need admission to hospital. This should be described.

Comments on the Quality of English Language

There are spelling mistakes littered throughout the paper. These should be corrected.

Author Response

Dear Editors and Reviewers,
Thank you for your letter and for the reviewers’ comments. Those comments are all valuable and very helpful for revising and improving our paper. We have studied comments carefully and have made correction which we hope meet with approval. Answers to the reviewer’s comments are as flowing:

Answers to Reviewer 2

 For the clinical paragraphs, we have emphasized more exhaustively the differences between CTLAT4/PD-1/PD-L1 induced hypophysitis. In this regard, a table has also been added (table 2). Figure 6 has been eliminated.